# Binge Drinking: The Top 100 Cited Papers

**DOI:** 10.3390/ijerph18179203

**Published:** 2021-08-31

**Authors:** María-Teresa Cortés-Tomás, José-Antonio Giménez-Costa, Beatriz Martín-del-Río, Consolación Gómez-Íñiguez, Ángel Solanes-Puchol

**Affiliations:** 1Basic Psychology Department, University of Valencia, 46010 Valencia, Spain; Maria.T.Cortes@uv.es; 2Department of Behavioural Sciences and Health, University Miguel Hernandez, 03205 Elche, Spain; bmartin@umh.es (B.M.-d.-R.); angel.solanes@goumh.umh.es (Á.S.-P.); 3Department of Basic and Clinical Psychology and Psychobiology, University Jaume I, 12071 Castellón de la Plana, Spain; iniguez@uji.es

**Keywords:** literature review, top 100, bibliometric analysis, binge drinking, alcohol abuse, drug use and health outcomes

## Abstract

We conducted a review to analyze the 100 most-cited studies on binge drinking (BD) in the Web of Science (WoS) database to determine their current status and the aspects that require further attention. We carried out a retrospective bibliometric analysis in January 2021. The year of publication, authors, design, subject, journal, institution and lead author’s country, as well as the definition of BD, were extracted from the articles. The data on the country, year, thematic category of the journals and their rank were obtained from the Institute for Scientific Information (ISI) Journal Citation Reports 2020. The number of citations was collected from the WoS, and the *h* index was collected from the Scopus database. The citation density and Bradford’s law were calculated. The majority of the articles were empirical quantitative studies with a cross-sectional design published between 1992 and 2013 in 49 journals. There were 306 authors, mostly English-speaking and from the USA. The definitions used to describe BD are not homogeneous. The most-cited topics were the analysis of consequences, determinants and epidemiology. There is a need to unify the definitions of BD and base them on scientific evidence. The multidisciplinary nature of BD is not well reflected in each of the thematic areas discussed in this work.

## 1. Introduction

Binge drinking (BD) is a risky alcohol consumption pattern with a high prevalence at the international level, especially among the youngest population [1,2,3,4].

Given that BD is a pattern of consumption with multiple consequences at the organizational, psychological and social levels for both the binge drinker and those around him or her [5,6,7], in recent years, much research has been generated on this topic.

One method that allows the quantitative analysis of scientific production and the thematic evolution of a determined field of research is bibliometry [8]. Metrics such as the number of publications, the number of citations or the impact factor (IF) of journals are often used as measures of relevance or productivity [9,10].

Using such metrics, the analysis of the most-cited articles on a topic allows us to identify which contributions are more visible, which have been more recognized or those that have exerted greater influence on the scientific community beyond the limits of their field of expertise [11,12,13]. Furthermore, from the point of view of science policy based on excellence, examining highly cited works has been considered an option for the detection and monitoring of “excellent” scientific research [11].

In recent years, numerous studies have been conducted to determine and analyze the most-cited papers in various fields of knowledge. In the health fields, such analyses have been performed in areas such as surgery [14], cardiovascular disease [15] and ophthalmology [16]. Analyses have also been conducted in areas of psychological health, such as occupational stress [17] and obsessive–compulsive disorder [18]. However, this methodological approach has been limited in the area of addictions [19]. The relevant studies have focused on substance use disorders [20,21], and bibliometric analyses on BD are nonexistent.

The main objective of this study is the analysis of the 100 most-cited articles in the Institute for Scientific Information (ISI) Web of Science (WoS) database (Thomson Reuters, Philadelphia, PA, USA) to provide a unique perspective on the current situation and opportunities for BD research, intervention and formation in the coming years.

## 2. Materials and Method

### 2.1. Search Methods

On 7 January 2021, the WoS Core Collection was searched with the keywords: “binge drinking”, “heavy episodic drinking”, “heavy drinking”, “heavy sessional drinking”, “dangerous drinking”, “risky single-occasion drinking”, “high-risk drinking”, “risky single occasion drinking”, “high risk drinking”, “excessive episodic consumption”, “frequent binge drinking”, “concentrated drinking episode” or “episodic heavy drinking”. These are all terms suggested by Cortés and Motos [1] in their review on how to define and measure BD. The search terms were included in the “topic” field (title, abstract, author’s keywords and KeyWords Plus). No filters were used in the fields of language, time, human studies, territory, affiliations or availability.

### 2.2. Search Outcome

A total of 15,894 studies were obtained. A modified approach to the method used by Lim et al. [22] and other authors [13,23,24] who have investigated the most-cited articles in different areas of healthcare was used. Three experts with more than 10 years of experience in BD research with youth (M.-T.C.-T., J.-A.G.-C. and C.G.-Í.) independently selected and analyzed the 100 articles with the highest number of citations. The inclusion criterion was that BD be the main topic or one of the main research variables. The exclusion criteria were studies that (1) were not related to alcohol consumption, (2) focused on general alcohol consumption without specific reference to BD or (3) considered BD as a secondary variable in the study. Disagreements among experts, although not frequent, were resolved in a consensus meeting. The flow of information through the different phases of the review (PRISMA flow diagram) is shown in Figure 1.

For articles with the same number of citations, the one with the highest citation density (total number of citations/years since publication) was classified first.

### 2.3. Quality Appraisal

An evaluation of the quality of the bibliographic sample was not performed since this study was a bibliometric analysis. As an objective indicator, we show the volume of citations in the WoS, which, although not an absolute measure of the quality of an article [25], is an index of the impact the article has exerted in the scientific community, with the assumption that high-quality research will result in a greater number of citations than lower-quality research [26].

### 2.4. Data Abstraction and Synthesis

Information was directly abstracted from the 100 articles for the year of publication, authors, design, subject, journal in which the article was published and the institution and country of the lead author. The design was classified into six categories [27]: (1) review studies, including systematic reviews and meta-analyses; (2) development and validation studies of instruments or scales; (3) cross-sectional studies, including follow-up studies using questionnaires, surveys or interviews; (4) papers using a qualitative methodology; (5) discussions on topics or methods; and (6) other types of studies that are not included in any of the other categories, e.g., longitudinal studies. The subject was also classified into six broad categories: (1) BD consequences, (2) determinants, (3) epidemiology, (4) intervention proposals, (5) consumption trajectories and (6) other, less frequent topics (evaluation, terminology discussions, typologies of consumers and psychiatric comorbidity). Finally, information was obtained on how BD was defined in each study according to three aspects [1]: amount of consumption, time interval and frequency.

From the ISI Journal Citation Reports 2020, data regarding the country of publication, the IF of the journals, its thematic category and its ranking in that category were obtained. The data on the number of citations for each study were obtained from the WoS, and the *h* index of the most-cited authors was obtained from the Scopus database.

Finally, the citation density of the articles and Bradford’s law were calculated. The citation density, or the average of annual citations, allowed us to obtain an index of the relative impact of an article regardless of the year of publication [23,28]. Bradford’s law [29,30] allowed the establishment of an objective measure of the weight of each journal in the top 100 by distributing the total number of journals into three productivity zones with a similar number of articles but a decreasing number of journals.

## 3. Results

### 3.1. Most-Cited Articles, Citation Density and Temporal Distribution

The top 100 most-cited articles related to BD are presented in Table 1 in descending order according to the number of citations. Together, they total 34,908 citations (ranging from 1280 to 180). Only 30 articles qualified as “citation classics”, i.e., having 400 or more citations [31].

All of the articles were published across a 21-year range, between 1992 and 2013 (shown in Figure 2). The period between 2000 and 2009 shows the highest number of studies (*n* = 79) and the highest number of citations (*n* = 27,181; 77.86%), and the period between 2010 and 2013 has the highest density of citations (34.52 citations/year).

### 3.2. Publication Type, Main Topics, BD Definition and Target Population

Most of the BD papers were empirical studies (*n* = 73). Of these, 39 were cross-sectional studies, including questionnaire and follow-up surveys or interviews, and 34 had a longitudinal design. Literature reviews were less frequent (*n* = 19). Of these, three were meta-analyses, nine were systematic reviews and seven reviews did not provide details on the methodology used to select the articles. Finally, of the remaining eight studies, two focused on the development or validation of an instrument or scale, one was a qualitative study and five were a discussion of a method or topic.

Regarding the topics of the 100 most important studies related to BD, the analysis of its consequences stands out, with 28 publications and 32.22% of the citations (*n* = 11,246). This topic also showed the greatest increase in citations since 1999 and began its decline very recently in 2016 (shown in Figure 3). The second most-cited topic (*n* = 27; 8784 citations, 25.17%) was the study of the variables or determinants underlying BD, with a stable and low volume of citations until 2004 and strong growth over the following 10 years.

With respect to the works on epidemiology (*n* = 14) and intervention proposals (*n* = 12), they had practically the same number of citations (*n* = 4324, 12.39%; *n* = 4034, 11.56%, respectively). The works on intervention were initially the second most cited between 2003 and 2006, with a large decrease since 2015. Finally, the works on the analysis of changes in consumption trajectories according to the growth and maturity of the subjects evaluated (*n* = 10; 3405 citations, 9.75%) increased their citations between 1996 and 2008 and have since remained stable. The rest of the articles focused on topics such as BD assessment (*n* = 3; 1384 citations), concept analysis (*n* = 3; 1121 citations), determination of consumer typologies (*n* = 2; 428 citations) or aspects related to psychiatric comorbidity (*n* = 1; 281 citations).

An analysis of the definitions of BD used (shown in Figure 4) showed that the most frequent (*n* = 48) differentiated the amount of alcohol consumption by gender (five or more standard drinks for men and four or more for women). In contrast, approximately 25% of the studies used a generic definition (five or more standard drinks) that did not differentiate by gender. The other articles offered specific definitions, and commonalities among them cannot be established. Three articles did not include a definition of BD.

Regarding the samples of these studies, interest in the university population stands out (*n* = 44). In addition, the USA population was the most frequently studied (*n* = 71).

### 3.3. Authors and Institutions

A total of 306 authors contributed to the 100 most-cited papers on BD. Five of the studies were single-authored [41,71,98,119,125], and 20 authors contributed to three or more articles (Table 2).

With regard to the authors’ affiliation, only the author who appears first in the author list was considered. These 100 authors belonged to 59 institutions from seven different countries, with a greater participation from the USA (*n* = 62), and 3 institutions from countries where English is not the first language (Finland, Lebanon and Switzerland).

Sixteen institutions contributed more than one paper. The institution with the largest contribution was the Harvard School of Public Health (USA) (*n* = 16). The University of California (USA) and the University of Washington (USA) appear second with five papers each. The Centers for Disease Control and Prevention (USA), Syracuse University (USA), the University of North Carolina at Chapel Hill (USA), the University of Michigan (USA) and the Center for Addiction and Mental Health at the University of Toronto (Canada) contributed three papers each.

### 3.4. Journals in Which the Top 100 Articles Were Published

A total of 49 journals published the 100 most-cited articles on BD (Table 3). After Bradford’s law was applied, the core distribution comprised three journals: the *Journal of Studies on Alcohol and Drugs* (known until 2006 as the *Journal of Studies on Alcohol*) (14 papers; 5322 citations); the *Journal of American College Health* (8 papers; 3373 citations) and *JAMA: Journal of the American Medical Association* (6 papers; 3074 citations).

As for the Journal Citation Reports (JCR) categories in which these journals are indexed (they can be classified in more than one category), 42.55% are in categories related to medicine (general, psychiatry, pediatrics, cardiology, obstetrics, etc.). Sixteen (30.04%) are in categories related to psychology (clinical, developmental and multidisciplinary); fourteen (29.78%) are in the category of substance abuse; and 23.40% are in the category of public, environmental and occupational health. The rest of the journals (*n* = 6) are classified into the categories of biochemistry, behavioral sciences, demography, education, neurosciences, neuroimaging, pharmacology and sport sciences.

The journals’ IFs ranged between 45.54 and 1.214 ($\bar{X}$ = 4.03), with the *JAMA*: *Journal of the American Medical Association* at the top. With respect to the JCR quartiles of these journals in their categories, 74.46% are located in Q1, 23.40% in Q2, 4% in Q3 and 1% in Q4. The rest (*n* = 2) have not had this value calculated since they were incorporated in their category this year. All but two of the journals are published in the USA (71.42%) or UK (24.48%).

## 4. Discussion

To our knowledge, this is the first bibliometric study to systematically identify and classify the top 100 most-cited papers in BD research. The results of our study provide various benefits for researchers and practitioners in BD. First, the results help new researchers understand the types of contributions, approaches, topics, populations and research methods applied in highly cited papers, enabling researchers to learn from them in writing higher-quality papers that will likely receive high numbers of citations. Second, such reviews help both veteran and new researchers identify topics with more visibility or impact so as to carry out more incremental research in those areas, as well as to identify topics on which more research should be conducted for greater visibility. Third, using such reviews, researchers and practitioners can identify the most frequently cited researchers and institutions with whom to collaborate, receive advice and training and so forth. Fourth, the results help practitioners identify research considered of higher scientific “excellence” in the specific field of BD and to use the techniques, programs or results reported in these studies.

The results show that these articles had 306 authors from 59 institutions in seven different countries. The country that contributed the most papers was the United States; moreover, most of the authors are from English-speaking institutions, as seen in similar studies [12,13,23,132]. Judging by these results, research in other countries that have this pattern of consumption needs greater visibility [19], as BD shows differences and similarities between countries that affect the generalization of the results [62].

In addition, the articles under study were published in 49 journals, of which only 14 belonged to the specific category on this topic of “substance abuse”, with the bulk of the journals belonging to the categories of medicine, public health and psychology. This differentiation reaffirms the multidisciplinary nature of the most visible research on BD, which transcends its specialty. If we focus on these journals’ IFs, as in other studies [13,133], the results confirm that the most-cited works on BD have been published mainly in journals located in the first and second quartiles of their categories. Furthermore, although the number of citations ranges between 1280 and 180, only 30 of these articles qualify as “citation classics” [31].

The topic most referred to in these works is that of specific medical consequences (e.g., cardiovascular effects, effects on the fetus and brain damage). The few works on the psychosocial consequences are often limited to listing them without exploring any in further detail.

The second most-cited topic is that of context-related BD determinants, specifically regarding the social norms of fellow drinkers or of the areas in which they live. Most are descriptive studies that do not estimate the weight or importance of these determinants in the development of BD and are also subject to uncontrolled cultural aspects. These characteristics make it difficult to generalize to other samples, other contexts or other countries.

The results of the review fail to reflect the multidisciplinary nature of BD in any topical area discussed in this work. In the studies on the consequences, physiological aspects predominate. The studies on determinants highlight those of a social nature. Regarding intervention, the most-cited articles are those of a psychosocial nature: brief motivational interventions, those focused on aspects of the consumer’s personality and, to a lesser extent, those based on normative feedback or marketing.

Several of the authors indicate that three main parameters are referenced in the research to characterize the operational definitions of BD [1,58]: the amount of alcohol consumed, the duration of the consumption episode and the time interval at which the presence of BD is recorded. In the most-cited work on BD, it is confirmed, on the one hand, that few articles include the three parameters and, on the other hand, that there is no consensus in defining each of these. Similar results have been found in recent reviews of the concept of BD [134].

The most commonly used parameter is the amount of alcohol ingested. However, in one in four studies, the definition of BD implemented does not align with any that are agreed upon in the literature. Additionally, very few differentiate the amount of alcohol ingested according to gender, ignoring the metabolic aspects associated with this substance [135,136]. Regarding the duration-of-consumption parameter, there is a notable inaccuracy of the interval evaluated, the most widely used being “a single occasion”. The low rigor with which these highly cited studies define the patterns of BD is striking. For BD, it is not only the large quantity of alcohol consumed that should be relevant but also that consumption is carried out in a short and quantifiable time interval, which could be indicated by the number of hours the person spent drinking [137].

The last of the parameters, the time interval of BD, accounts for the episodic and irregular nature of this consumption pattern. The time periods defined by many of these works are short and do not allow accounting for this discontinuity, such as “the past week” or “the last 30 days”.

Therefore, the diversity of the definitions found in the most-cited articles on BD, with special mention of nondifferentiation by gender or the differences in the time intervals of consumption, highlights the need to visualize conceptual advances in BD that will allow the homogenization of the results derived from the research.

As limitations of this work, first, we used only one database for our search for the most-cited papers. This procedure conditioned our list and ranking, which may have differed if we had used a different database, such as Scopus or Google Scholar [138]. Citations obtained exclusively from the WoS are limited to the sources selected using this database, which, for example, does not include citations in books and journals published in languages other than English. Nevertheless, we chose this database because, although it does not provide complete coverage, it has been extensive and multidisciplinary since 1900 and includes more than 12,000 journals with an impact across the world, including open access journals [139]. Second, self-citations and citations in lectures, textbooks and web-based literature have not been considered [140]. The positive or negative nature of the citations of the top 100 articles were not considered in this work, and thus, we were unable to determine the level of agreement, disagreement or criticism from the scientific community regarding the topics covered in these articles [141,142].

## 5. Conclusions

The analysis of the 100 most-cited articles in BD offers a unique perspective on the most visible works in this area, providing insights about the need to unify the definitions of binge drinking based on the scientific evidence agreed upon by experts. Moreover, the topical areas analyzed do not clearly reflect the multidisciplinary nature of the BD. In the same way, research from other countries that have this pattern of consumption needs greater visibility to show the differences and similarities between countries that will affect the generalization of the results.

## Figures and Tables

**Figure 1 ijerph-18-09203-f001:**
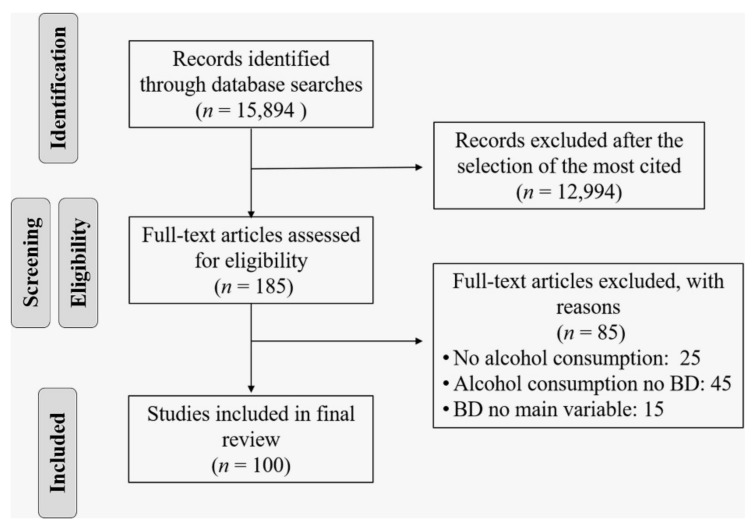
PRISMA flow diagram.

**Figure 2 ijerph-18-09203-f002:**
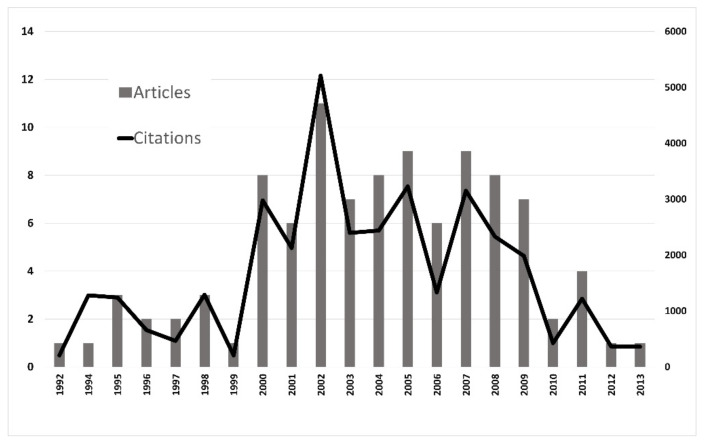
Number of articles and citations.

**Figure 3 ijerph-18-09203-f003:**
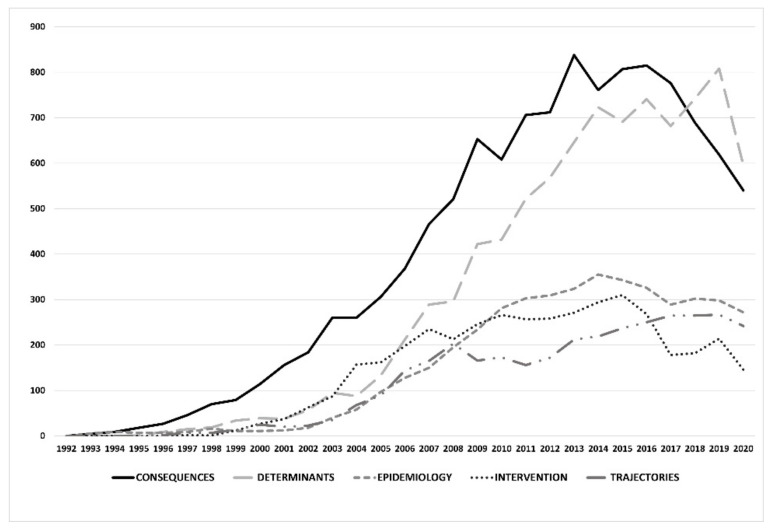
Temporal evolution of citations according to the topic of the 100 most-cited papers in BD.

**Figure 4 ijerph-18-09203-f004:**
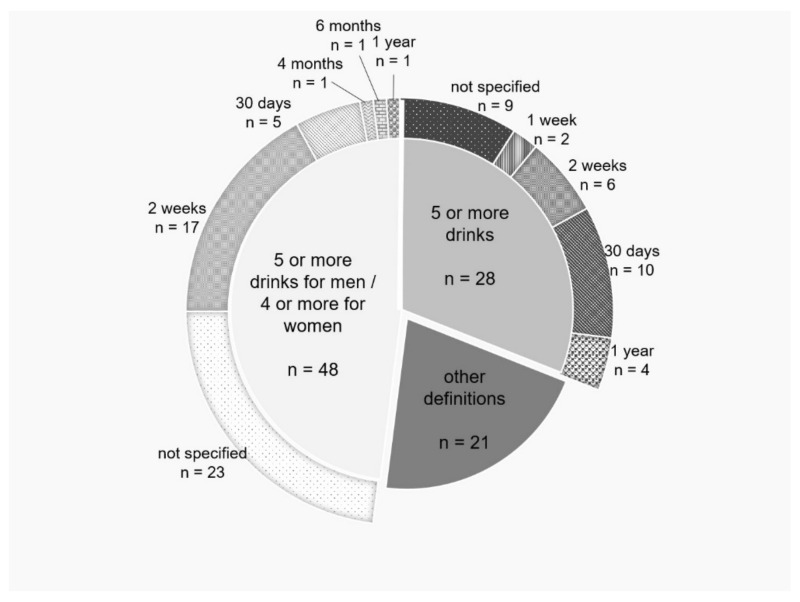
BD definitions in the top 100 papers.

**Table 1 ijerph-18-09203-t001:** The 100 most-cited papers about binge drinking in the WoS.

Rank	Cited	Citation Density ^1^	Author(s) and Year of Publication	First Author’s Institution and Country	Type of Article *^2^*
**1**	1280	49.23	Wechsler et al. (1994) [32]	Harvard Sch. of Public Health (USA)	3
**2**	1064	59.11	Wechsler et al. (2002a) [33]	Harvard Sch. of Public Health (USA)	6
**3**	807	44.83	O’Malley and Johnston (2002) [34]	Univ. of Michigan (USA)	6
**4**	802	53.47	Kuntsche et al. (2005) [35]	Swiss Inst. for the Prevention of Alcohol and Drug Problems (Switzerland)	1
**5**	798	42.00	Borsari and Carey (2001) [36]	Syracuse Univ. (USA)	1
**6**	748	37.40	Wechsler et al. (2000) [37]	Harvard Sch. of Public Health (USA)	6
**7**	742	49.47	Hingson et al. (2005) [38]	Boston Univ. Sch. of Public Health (USA)	6
**8**	680	40.00	Naimi et al. (2003) [39]	Centers for Disease, Control and Prevention (USA)	6
**9**	637	28.95	Marlatt et al. (1998) [40]	Univ. of Washington (USA)	6
**10**	586	32.56	Perkins (2002) [41]	Hobart and William Smith Colleges (USA)	1
**11**	577	44.38	Crews et al. (2007) [42]	Univ. of North Carolina at Chapel Hill (USA)	1
**12**	569	31.61	Schulenberg and Maggs (2002) [43]	Univ. of Michigan (USA)	1
**13**	556	42.77	Miller et al. (2007) [44]	Center for Disease Control and Prevention (USA)	3
**14**	529	58.78	Bouchery et al. (2011) [45]	Mathematica Policy Research (USA)	3
**15**	514	20.56	Wechsler et al. (1995b) [46]	Harvard Sch. of Public Health (USA)	3
**16**	512	20.48	Wechsler et al. (1995a) [47]	Harvard Sch. of Public Health (USA)	3
**17**	503	27.94	Chassin et al. (2002) [48]	Arizona State Univ. (USA)	6
**18**	478	36.77	Carey et al. (2007) [49]	Syracuse Univ. (USA)	1
**19**	463	21.05	Wechsler et al. (1998) [50]	Harvard Sch. of Public Health (USA)	6
**20**	457	22.85	Wilsnack et al. (2000) [51]	Univ. of North Dakota (USA)	3
**21**	442	27.63	Neighbors et al. (2004) [52]	North Dakota State Univ. (USA)	6
**22**	439	25.82	Ham and Hope (2003) [53]	Univ. of Nebraska–Lincoln (USA)	5
**23**	439	24.39	Knight et al. (2002) [54]	Harvard Medical Sch./Children’s Hosp. (USA)	3
**24**	431	25.35	Bradley et al. (2003) [55]	Health Services Research and Development Service (USA)	2
**25**	427	17.79	Schulenberg et al. (1996) [56]	Univ. of Michigan (USA)	6
**26**	420	35.00	Jacobson et al. (2008) [57]	Naval Health Research Center (USA)	6
**27**	417	37.91	Courtney and Polich (2009) [58]	San Diego State Univ. (USA)	1
**28**	415	31.92	Neighbors et al. (2007) [59]	Univ. of Washington (USA)	3
**29**	414	20.70	Borsari and Carey (2000) [60]	Syracuse Univ. (USA)	3
**30**	408	21.47	Wechsler and Nelson (2001) [61]	Harvard Sch. of Public Health (USA)	4
**31**	395	24.69	Kuntsche et al. (2004) [62]	Swiss Inst. for the Prevention of Alcohol and Drug Problems (Switzerland)	1
**32**	387	35.18	Crews and Boettiger (2009) [63]	Univ. of North Carolina at Chapel Hill (USA)	5
**33**	369	46.13	Chen and Jacobson (2012) [64]	Univ. of Chicago (USA)	6
**34**	366	20.33	Fleming et al. (2002) [65]	Univ. of Wisconsin–Madison Medical Sch. (USA)	6
**35**	363	51.86	White and Hingson (2013) [66]	Duke Univ. Medical Center (USA)	1
**36**	361	30.08	Brown et al. (2008) [67]	Univ. of California (USA)	5
**37**	350	17.50	Crews et al. (2000) [68]	Univ. of North Carolina at Chapel Hill (USA)	3
**38**	332	17.47	Baer et al. (2001) [69]	Univ. of Washington (USA)	6
**39**	328	29.82	Wilsnack et al. (2009) [70]	Univ. of North Dakota (USA)	3
**40**	319	21.27	Slutske (2005) [71]	Univ. of Missouri–Columbia (USA)	3
**41**	319	19.94	Dawson et al. (2004) [72]	U.S. Department of Health and Human Services (USA)	3
**42**	319	15.95	Hill et al. (2000) [73]	Univ. of Washington/Rutgers Univ. (USA)	6
**43**	313	19.56	Chassin et al. (2004) [74]	Arizona State Univ. (USA)	6
**44**	299	24.92	Wechsler and Nelson (2008) [75]	Harvard Sch. of Public Health (USA)	6
**45**	299	24.92	O’Brien et al. (2008) [76]	Wake Forest Univ. Sch. of Medicine (USA)	3
**46**	296	18.50	Del Boca et al. (2004) [77]	Univ. of South Florida (USA)	6
**47**	295	24.58	Keyes et al. (2008) [78]	New York State Psychiatric Inst./Columbia Univ. (USA)	6
**48**	282	15.67	Andrews et al. (2002) [79]	Oregon Research Inst. (USA)	6
**49**	281	23.42	Strine et al. (2008) [80]	Centers for Disease Control and Prevention (USA)	3
**50**	281	21.62	O’Keefe et al. (2007) [81]	Univ. of Missouri–Kansas City Sch. of Medicine (USA)	1
**51**	278	12.09	Douglas et al. (1997) [82]	U.S. Department of Health and Human Services (USA)	3
**52**	274	19.57	White et al. (2006) [83]	The State Univ. of New Jersey (USA)	3
**53**	265	20.38	Borsari et al. (2007) [84]	Brown Univ. (USA)	1
**54**	262	23.82	Squeglia et al. (2009) [85]	Univ. of California (USA)	5
**55**	258	12.90	Holder et al. (2000) [86]	Pacific Inst. for Research and Evaluation, Berkeley (USA)	6
**56**	257	16.06	Mohler-Kuo et al. (2004) [87]	Harvard Sch. of Public Health (USA)	3
**57**	255	17.00	Zeigler et al. (2005) [88]	American Medical Association (USA)	1
**58**	248	12.40	Muthén and Muthén (2000) [89]	Univ. of California (USA)	6
**59**	246	14.47	Wechsler et al. (2003) [90]	Harvard Sch. of Public Health (USA)	3
**60**	244	16.27	Pitkänen et al. (2005) [91]	Univ. of Jyväskylä (Finland)	6
**61**	240	16.00	Martens et al. (2005) [92]	Univ. at Albany (USA)	2
**62**	238	23.80	Elder et al. (2010) [93]	National Center for Health Marketing (USA)	1
**63**	237	26.33	Patra et al. (2011) [94]	Centre for Addiction and Mental Health/Univ. of Toronto (Canada)	1
**64**	235	16.79	Harris et al. (2006) [95]	The Univ. of North Carolina at Chapel Hill (USA)	6
**65**	232	9.67	Haines and Spear (1996) [96]	Northern Illinois Univ. (USA)	6
**66**	230	25.56	Chen et al. (2011) [97]	Brigham and Women’s Hosp. and Harvard (USA)	6
**67**	225	14.06	Jennison (2004) [98]	Univ. of Northern Colorado (USA)	6
**68**	223	24.78	King, et al. (2011) [99]	The Univ. of Chicago (USA)	3
**69**	221	15.79	Bell et al. (2006) [100]	Indiana Univ. Sch. of Medicine (USA)	3
**70**	220	14.67	Jaccard et al. (2005) [101]	Florida International Univ. (USA)	3
**71**	215	8.60	Agostinelli et al. (1995) [102]	Univ. of New Mexico (USA)	3
**72**	213	14.20	Townshend and Duka (2005) [103]	Univ. of Sussex (UK)	3
**73**	212	12.47	Weitzman et al. (2003) [104]	Harvard Sch of Public Health (USA)	3
**74**	210	15.00	Conrod et al. (2006) [105]	Univ. of London (UK)	3
**75**	209	12.29	Johnston and White (2003) [106]	Queensland Univ. of Technology (Australia)	6
**76**	209	11.61	Wechsler et al. (2002b) [107]	Harvard Sch. of Public Health (USA)	3
**77**	207	14.79	White et al. (2006) [108]	Duke Univ. Medical Center (USA)	3
**78**	206	9.81	Puddey et al. (1999) [109]	Univ. of Western Australia (Australia)	1
**79**	206	7.36	Wechsler and Isaac (1992) [110]	Harvard Sch. of Public Health (USA)	3
**80**	205	15.77	Popova et al. (2007) [111]	Centre for Addiction and Mental Health/Univ. of Toronto (Canada)	3
**81**	204	18.55	Hingson and Zha (2009) [112]	National Inst. on Alcohol Abuse and Alcoholism (USA)	6
**82**	200	10.53	Nelson and Wechsler (2001) [113]	Harvard Sch. of Public Health (USA)	3
**83**	198	11.00	Guo et al. (2002) [114]	Univ. of Washington (USA)	6
**84**	197	17.91	Ethen et al. (2009) [115]	Texas Department of State Health Services (USA)	3
**85**	196	16.33	Szmigin et al. (2008) [116]	Univ. of Birmingham (UK)	5
**86**	196	10.32	Wood et al. (2001) [117]	Univ. of Rhode Island (USA)	3
**87**	194	12.93	Tucker et al. (2005) [118]	The RAND Corporation (USA)	6
**88**	194	12.13	Weitzman (2004) [119]	Harvard Sch. of Public Health (USA)	3
**89**	194	10.21	Maier and West (2001) [120]	The Texas A&M Univ. System Health Science Center (USA)	1
**90**	192	8.73	Leichliter et al. (1998) [121]	Southern Illinois Univ. Carbondale (SIUC) (USA)	3
**91**	189	18.90	Roerecke and Rehm (2010) [122]	Centre for Addiction and Mental Health (Canada)	1
**92**	189	17.18	Blazer and Wu (2009) [123]	Duke Univ. Medical Center (USA)	3
**93**	189	14.54	Viner and Taylor (2007) [124]	Univ. College Hosp.	6
**94**	187	10.39	Gill (2002) [125]	Queen Margaret Univ. College	1
**95**	187	8.13	Wechsler et al. (1997) [126]	Harvard Sch. of Public Health (USA)	3
**96**	185	14.23	Karam et al. (2007) [127]	St George Hosp. Univ. Medical Center/Balamand Univ./Inst. for Development Research Advocacy and Applied Care (Lebanon)	1
**97**	185	9.25	Bensley et al. (2000) [128]	Washington State Department of Health (USA)	3
**98**	183	13.07	Duncan et al. (2006) [129]	Northwestern Univ. (USA)	3
**99**	181	15.08	Conrod et al. (2008) [130]	King’s College (UK)	6
**100**	180	10.59	Tucker et al. (2003) [131]	Univ. of California/RAND (USA)	6

^1^ Citation density: mean number of citations per year. ^2^ Type of article: (1) review: including literature and systematic review, and meta-analysis; (2) instrument validation: including development or validation of a psychometric instrument or scale; (3) cross-sectional study: including questionnaire and follow-up surveys, or interviews; (4) qualitative study or methods for qualitative study; (5) discussion, including discussion of a method or topic; (6) longitudinal study.

**Table 2 ijerph-18-09203-t002:** Authors who contributed to the most papers among the top 100.

Author	Institution	Freq.	Scopus Index
Henry Wechsler	Dept. of Health and Social Behavior, Harvard Sch. of Public Health, Boston, USA	17	67
Toben F. Nelson	Dept. of Health and Social Behavior, Harvard Sch. of Public Health, Boston, USA	7	33
George W. Dowdall	Dept. of Health and Social Behavior, Harvard Sch. of Public Health, Boston, USA/ Dept. of Sociology, St. Joseph’s Univ. Hosp., Philadelphia, USA/ Dept. of Community Health, Brown Univ. Sch. of Medicine, Providence, USA	6	11
Meichun Kuo /Meichun Mohler-Kuo	Dept. of Health and Social Behavior, Sch. of Public Health, Boston, USA/ Hosp. Boston, USA	5	24
Robert D. Brewer	Analytic Methods Branch, Centers for Disease Control and Prevention (CDC), Atlanta, Georgia, USA/ National Center for Chronic Disease Prevention and Health Promotion, Centers for Disease Control and Prevention (CDC), Atlanta, USA	4	41
Andrea E. Davenport	Dept. of Health and Social Behavior, Harvard Sch. of Public Health, Boston, USA	4	6
Ralph W. Hingson	Division of Epidemiology and Prevention Research, National Institute on Alcohol Abuse and Alcoholism, Bethesda, USA/ Boston Univ. Sch. of Public Health, Center to Prevent Alcohol Problems Among Young People, Boston, USA	4	54
Jae Eun Lee	Dept. of Health and Social Behavior, Harvard Sch. of Public Health, Boston, USA/ Univ. of Nevada, Las Vegas, USA	4	13
Jürgen Rehm	Social and Epidemiological Research Dept. (SER), Centre for Addiction and Mental Health, Toronto, Canada/ Research Institute for Public Health and Addiction, Zurich, Switzerland/ Klinische Psychologie und Psychotherapie, Technische Universität Dresden, Germany/ Dalla Lana Sch. of Public Health, Univ. of Toronto, Canada/ Dept. of Psychiatry, Univ. of Toronto, Canada	4	108
Brian E. Borsari	Center for Health and Behavior, Syracuse Univ., New York, USA/ Dept. of Psychology, Syracuse Univ., New York, USA	3	35
Kate B. Carey	Center for Health and Behavior, Syracuse Univ., New York, USA/ Dept. of Psychology, Syracuse Univ., New York, USA	3	70
Richard F. Catalano	Social Development Research Group, Sch. of Social Work, Univ. of Washington, Seattle, USA	3	78
Fulton T. Crews	Bowles Center for Alcohol Studies, Univ. of North Carolina at Chapel Hill, USA/ Department of Pharmacology, Univ. of North Carolina at Chapel Hill, USA/ Department of Psychiatry, Univ. of North Carolina at Chapel Hill, USA	3	68
Gerhard Gmel	Research Dept., Swiss Institute for the Prevention of Alcohol and Drug Problems (SIPA), Switzerland/ Alcohol Treatment Center, Lausanne Univ. Hosp., Switzerland	3	58
Mary E. Larimer	Dept. of Psychology, Univ. of Washington, USA	3	60
Hang Lee	Dept. of Health and Social Behavior, Harvard Sch. of Public Health, Boston, USA/ Massachusetts General Hosp., Boston, USA/ Center for Vaccine Research at the Univ. of California, Los Angeles, USA/ Center for Vaccine Research and Dept. of Pediatrics at the Univ. of California, Los Angeles Sch. of Medicine in Torrance, USA	3	64
Timothy S. Naimi	Alcohol Team, Emerging Investigations, and Analytics Methods Research. Centers for Disease, Control and Prevention, USA	3	43
John E. Schulenberg	Survey Research Center, Institute for Social Research, Univ. of Michigan, Ann Arbor, Michigan, USA/ Dept. of Psychology, Univ. of Michigan, USA	3	56
Mark Seibring	Dept. of Health and Social Behavior. Harvard Sch. of Public Health in Boston, USA	3	8
Elissa R. Weitzman	Dept. of Health and Social Behavior, Harvard Sch. of Public Health, Boston, USA	3	26

**Table 3 ijerph-18-09203-t003:** Journals with the most productive areas according to Bradford’s law.

			Journal Citation Reports (JCR) 2019			
Journal	F	Citations	IF ^1^	Ranking SCIE ^2^	Ranking SSCI ^3^	Founding Year
*J. of Studies on Alcohol* (*n* = 12) (until 2006) + *J. of Studies on Alcohol and Drugs* (*n* = 2)	14	5322	2.448	12/20SA	16/36SA	(1940) 1975/2007
*J. of American College Health*	8	3373	1.710	-	119/263ER 84/171PEOH	1982
*JAMA: The J. of the American Medical Association*	6	3074	45.540	3/165MGI	-	1960
*J. of Consulting and Clinical Psychology*	5	2292	4.632	-	10/31PC	1968
*Alcoholism: Clinical and Experimental Research*	5	1455	3.035	8/20SA	-	1977
*Addiction*	4	1234	6.343	2/20SA 15/155PS	2/36SA 11/142PS	1993
*American J. of Public Health*	3	1358	6.464	13/193PEOH	7/171PEOH	1971
*American J. of Preventive Medicine*	3	952	4.420	24/193PEOH 25/165MGI	9/171PEOH	1985
*J. of Adolescent Health*	3	779	3.945	30/193PEOH 8/128PD	14/171PEOH 7/77 PD	1971

^1 ^ IF: Impact factor according to the JCR Science Citation Index Expanded (SCIE) and Social Sciences Citation Index (SSCI) 2019. ^2^ JCR ranking in the SCIE (SA: substance abuse; MGI: medicine, general and internal; PS: psychiatry; PEOH: public, environmental and occupational health; P: pediatrics). ^3^ JCR ranking in the SSCI (SA: substance abuse; ER: education and educational research; PEOH: public, environmental and occupational health; PC: psychology, clinical; PS: psychiatry; PD: psychology, developmental).

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
