# Peer review of "Binge Drinking: The Top 100 Cited Papers"

_ijerph, 2021, doi:10.3390/ijerph18179203_

Round 1
Reviewer 1 Report
There is a well done analysis of the observations binge eating. The discussion is comprehensive and deep, showing the authors' great erudition in this field. The conclusions were correctly drawn from the results. References are up to date and properly selected. However, I suggest a few correct: 1. I would like to know which complete definition of Binge Eating was used by the authors or if they used from the DSM-5 classification? (American Psychiatric Association. Feeding and Eating Disorders. In: Diagnostic and Statistical Manual of Mental Disorders (DSM-5). 5th ed. American Psychiatric Publishing, Washington, DC, London, England. 2013; or https://icd.who.int/browse11/l-m/en; ICD-11 for Mortality and Morbidity Statistics (Version: 04/2019), 2019; or maybe even https: /icd.who.int/browse10/2016/en#/F50, ICD-10 Version: 2016). 2. I suggest using the PRISMA scheme in the methodology, which will better illustrate the research methodology better and be more understandable for readers.Please consider updating your review layout according to PRISMA Statement (the Preferred Reporting Items for Systematic Reviews and Meta-Analyses) - the REVIEW should be divided according to the guidelines available from www.prisma-statement.org
Please prepare a PRISMA flow diagram and describe your search strategy to help the readers understand your choice of literature - this may be added as your additional Figure.
A PRISMA flow diagram allows to see what literature you have searched through to compose your review.
3. Why did the authors only mention comments about the USA or Europe in the summary, not taking into account other high consumption countries (Russia, Ukraine, Belarus, Scandinavian countries, or Australia or New Zealand? 4. Please explain to future readers what kind of alcohol was taken into account for the analysis, as well as its frequency and quality. 5. Figure 4 shows n = 97 (48 + 28 + 21), please explain this. Is that correct? 6. So what interventions should be undertaken or even the methods of observation should be maintained while maintaining the standards for the largest regions of the world since the authors have already touched on the comparison between the USA and Europe. To sum up, after explaining the strengths and weaknesses in the text, I recommend publishing the article after the changes.Author Response
First of all, thank you for your review and for your suggestions in relation to our work.
Regarding these suggestions:
1. We understand that when the reviewer refers to Binge Eating he is actually referring to Binge Drinking.
A priori we do not start from any one definition, but use the terms suggested by Cortés and Motos in their review on how to define and measure BD. These terms were used to carry out the search for papers. Figure 4 shows the definitions extracted from the review of these papers.
2. Following the reviewer's recommendation, the title of Figure 1 has been renamed to better fit its content. Thus, both the headings of this section and the figure have been adapted to PRISMA terminology.
3. We are grateful for the reviewer's suggestion and, accordingly, the lines referring to the need to give greater visibility in research to other countries beyond Europe have been modified both in the discussion and in the conclusions.
4. In the articles included in this review, the type of alcohol is not specified, since in most of the studies consumption is recorded in Standard Drinking Units (SDU), which are specific to each country.
5. An annotation has been included in the text referring to Figure 4 to correct the error indicated by the reviewer. ("Three articles do not include a definition of BD").
6. As indicated in the comment on point 3, some parts of the discussion and conclusions have been modified to take into account the international representation of BD. But we thank you for the comment included because it would be of great interest for the advancement of this issue to make a comparison between countries using a different methodology than the Top 100 used in this study.
Reviewer 2 Report
The structure and methodology of the article is well organized. The number of articles consulted is sufficient to guarantee good quality and the scientific consistency of the work. The graphs enrich the theoretical component.
No critical issues are highlighted. A single suggestion concerns the inclusion of possible hypotheses of further future works, given the social weight of the issue addressed.
Author Response
Thank you very much for your review. We have made some minor changes to the manuscript.
Reviewer 3 Report
The authors of this work pose an interesting review on the concept of binge drinking and the need for a standardized definition. The authors found that there is no unity on the term across 100 highly cited papers on binge drinking. While there is no unity, the authors might want to comment on a summary metric of BD based on what they found in these 100 publications and what they would recommend after such a review.
Below are some other noted comments/uggestions:
1. Define what the 'type of article' numbers in table 1 refer to in the table legend if there is space.
2. A figure highlighting the differences in BD definitions across countries would be helpful to the reader and might be a new piece of information to the literature of this type of review.
3. There is a lot of discussion and analysis on citations and citation numbers but not a lot about the topic of BD itself. Please add a bit more about the concept of BD in the results.
4. In Figure 4, there is a section for other definitions n=21, perhaps this can be put into a table or supplement with some other data about them. This might be interesting since the common measure is >4-5 drinks per day.
4. Might be useful to see a distribution on alcohol types consumed reported in the articles, if any, i.e. beer, liquor, wine, etc.
Author Response
First of all, thank you for your review and for your suggestions in relation to our work.
The first comment proposed by the reviewer is in the discussion (“Several authors indicate that three main parameters are referenced in research to characterize operational definitions of BD [1,58]: the amount of alcohol consumed, the duration of the consumption episode and the time interval at which the presence of BD is recorded. In the most cited work on BD, it is confirmed, on the one hand, that few articles include the three parameters, and on the other hand, that there is no consensus in defining each of these. Similar results have been found in recent reviews of the concept of BD [136]. The most commonly used parameter is the amount of alcohol ingested. However, in one in four studies, the definition of BD implemented does not align with any that are agreed upon in the literature. Additionally, very few differentiate the amount of alcohol ingested according to gender, ignoring the metabolic aspects associated with this substance [137,138]. Regarding the duration-of-consumption parameter, there is a notable inaccuracy of the interval evaluated, the most widely used being “a single occasion”. The low rigor with which these highly cited studies define patterns of BD is striking. For BD, it is not only the large quantity of alcohol consumed that should be relevant but also that consumption is carried out in a short and quantifiable time interval, which could be indicated by number of hours the person spent drinking [139]. The last of the parameters, the time interval of BD, accounts for the episodic and irregular nature of this consumption pattern. The time periods defined by many of these works are short and do not allow accounting for this discontinuity, such as “the past week” or “the last 30 days”. Therefore, the diversity of definitions found in the most cited articles on BD, with special mention of nondifferentiation by gender or differences in the time intervals of consumption, highlights the need to visualize conceptual advances in BD that will allow the homogenization of the results derived from the research.”).
1. Thank you for pointing out the error. It has been corrected by including the corresponding notes to Tables 1 and 3.
2. This differentiation has not been included due to the overrepresentation of US research.
3. We have considered it appropriate to give greater presence to the concept of BD in the discussion, given that in the results section Figure 4, comprising the data from the 100 articles, includes all the objective numerical information that can be extracted on the concept of BD.
4. It has not been considered convenient to include this information given the heterogeneity and lack of precision of these definitions. In this group, definitions such as "until very drunk" are included.
5. In most of the articles included in this selection, this information does not appear directly because country-specific Standard Drinking Units (SDU) are recorded.
Round 2
Reviewer 1 Report
Well done!
The manuscript is well written.